# Learning in Generalized Linear Contextual Bandits with Stochastic Delays

Zhengyuan Zhou[1,2]* Renyuan Xu[3]* and Jose Blanchet[4]
[1] Department of Electrical Engineering, Stanford University
[2] Bytedance Inc.
[3] Department of Industrial Engineering and Operations Research, UC Berkeley
[4] Department of Management Science and Engineering, Stanford University

## Abstract

In this paper, we consider online learning in generalized linear contextual bandits where rewards are not immediately observed. Instead, rewards are available to the decision maker only after some delay, which is unknown and stochastic, even though a decision must be made at each time step for an incoming set of contexts. We study the performance of upper confidence bound (UCB) based algorithms adapted to this delayed setting. In particular, we design a delay-adaptive algorithm, which we call Delayed UCB, for generalized linear contextual bandits using UCB-style exploration and establish regret bounds under various delay assumptions. In the important special case of linear contextual bandits, we further modify this algorithm and establish a tighter regret bound under the same delay assumptions. Our results contribute to the broad landscape of contextual bandits literature by establishing that UCB algorithms, which are widely deployed in modern recommendation engines, can be made robust to delays.

## 1  Introduction

The growing availability of user-specific data has welcomed the exciting era of personalized recommendation, a paradigm that uncovers the heterogeneity across individuals and provides tailored service decisions that lead to improved outcomes. Such heterogeneity is ubiquitous across a variety of application domains (including online advertising, medical treatment assignment, product/news recommendation (Li et al. (2010), Bubeck et al. (2012),Chapelle (2014),Bastani and Bayati (2015),Schwartz et al. (2017))) and manifests itself as different individuals responding differently to the recommended items. Rising to this opportunity, contextual bandits have emerged to be the predominant mathematical formalism that provides an elegant and powerful formulation: its three core components, the features (representing individual characteristics), the actions (representing the recommendation), and the rewards (representing the observed feedback), capture the salient aspects of the problem and provide fertile ground for developing algorithms that balance exploring and exploiting users' heterogeneity.

As such, the last decade has witnessed extensive research efforts in developing effective and efficient contextual bandits algorithms. In particular, two types of algorithms–upper confidence bounds (UCB) based algorithms (Li et al. (2010); Filippi et al. (2010); Chu et al. (2011); Jun et al. (2017); Li et al. (2017)) and Thompson sampling (TS) based algorithms (Agrawal and Goyal (2013a,b); Russo and Van Roy (2014, 2016); Abeille et al. (2017))–stand out from this flourishing and fruitful line of work: their theoretical guarantees have been analyzed in many settings, often yielding (near-)optimal regret bounds; their empirical performance have been thoroughly validated, often providing insights into

their practical efficacy (including the consensus understanding that TS-based algorithms often suffer from intensive computation for posterior updates but can leverage a correctly specified prior and have superior empirical performance; UCB-based algorithms can often achieve tight theoretical regret bounds but are often sensitive to hyper-parameter tuning in empirical performance). To a large extent, these two family of algorithms have been widely deployed in many modern recommendation engines.

However, a key assumption therein–both the algorithm design and their analyses–is that the reward is immediately available after an action is taken. Although useful as a first-step abstraction, this is a stringent requirement that is rarely satisfied in practice, particularly in large-scale systems where the time-scale of a single recommendation is significantly smaller than the time-scale of a user's feedback. For instance, in E-commerce, a recommendation is typically made by the engine in milliseconds, whereas a user's response time (i.e. to buy a product or conversion) is typically much larger, ranging from hours to days, sometimes even to weeks. Similarly, in clinical trials, it is infeasible to immediately observe and hence take into account the medical outcome after applying a treatment to a patient–collecting medical feedback can be a time-consuming and often random process; and in general, it is common to have applied trial treatments to a large number of patients, with individual medical outcomes only available much later at different, random points in time. In both the E-commerce (Kannan et al. (2001); Chapelle (2014); Vernade et al. (2017))and the clinical trials cases (Chow and Chang (2011)), a random and often significantly delayed reward is present, thereby requiring adjustments in classical formulations to understand the impact of delays.

## 1.1 Related Work

The problem of learning on bandits with delays has recently been studied in different settings in the existing literature, where most of the efforts have concentrated on the multi-armed bandits setting, including both the stochastic and the adversarial multi-armed bandits. For stochastic multi-armed bandits with delays, Joulani et al. (2013) show a regret bound $O(\log T + \mathbb{E}[\tau] + \sqrt{\log T \mathbb{E}[\tau]})$ where $\mathbb{E}[\tau]$ is the mean of the **iid** delays. Desautels et al. (2014) consider Gaussian Process bandits with a bounded stochastic delay. Mandel et al. (2015) follow the work of Joulani et al. (2013) and propose a queue-based multi-armed bandit algorithm to handle delays. Pike-Burke et al. (2017) match the same regret bound as in Joulani et al. (2013) when feedback is not only delayed but also anonymous.

For adversarial multi-armed bandits with delays, Neu et al. (2010) establish the regret bound of $\mathbb{E}[R_T] \leq O(\tau_{\text{const}}) \times \mathbb{E}[R'_T(\frac{T}{\tau_{\text{const}}})]$ for Markov decision process, where $\tau_{\text{const}}$ is the constant delay and $R'_T$ is the regret without delays. Cesa-Bianchi et al. (2019) consider adversarial bandits with fixed constant delays on the network graph, with a minimax regret of the order $\tilde{O}\left(\sqrt{(K + \tau_{\text{const}})T}\right)$, where $K$ is the number of arms. Another related line of work is adversarial learning with full information (all arms' rewards are observed), where its different variants in the delayed setting have been studied by Weinberger and Ordentlich (2002), Mesterharm (2005), Quanrud and Khashabi (2015) and Garrabrant et al. (2016). Very recently, Bistritz et al. (2019) studied adversarial bandits learning under arbitrary delays using Exp3 and established finite-sample delay-adaptive regret bounds.

On the other hand, learning in contextual bandits with delays are much less explored. Joulani et al. (2013) consider learning on adversarial contextual bandits with delays and establish an expected regret bound $\mathbb{E}[R_T] \leq (1 + \mathbb{E}[M_T^*]) \times \mathbb{E}\left[R'_T\left(\frac{T}{1+\mathbb{E}[M_T^*]}\right)\right]$ by using a black-box algorithm, where $M_T^*$ is the running maximum number of delays up to round $T$. Dudik et al. (2011) consider stochastic contextual bandits with a fixed constant delay. The reward model they consider is general (i.e. not necessarily parametric); however, they require the policy class to be finite. In particular, they obtain the regret bound $O(\sqrt{K \log N}(\tau_{\text{const}} + \sqrt{T}))$, where $N$ is the number of policies and $\tau_{\text{const}}$ is again the fixed constant delay. On a related front, Grover et al. (2018b) studied the problem of best-arm identification under delayed feedback. There, the objective is to identify the best arm using as few samples as possible, without taking into account the cost incurred along the way (i.e. a different objective from regret). In closing, we also mention that there is also a growing literature in offline contextual bandits learning Swaminathan and Joachims (2015); Kitagawa and Tetenov (2018); Zhou et al. (2018a). In this domain, delay is typically not a concern as all the data has already been collected in a single batch before any learning/decision-making takes place.

## 1.2 Our Contributions

In this paper, we consider learning on generalized linear (stochastic) contextual bandits with stochastic delays. More specifically, we design a delay-adaptive algorithm for generalized linear contextual bandits using UCB-style exploration, which we call Delayed UCB (DUCB, as given in Algorithm 1). DUCB requires a carefully designed delay-adaptive confidence parameter, which depends on how many rewards are missing up to the current time step. Next, we give regret characterizations of DUCB under independent stochastic, unbounded delays. In particular, as a special case of our results, when the delays are **iid** with mean $\mu_D$, we establish a high-probability regret bound of $\tilde{O}\left(\left(\sqrt{\mu_D d} + \sqrt{\sigma_G d} + d\right)\sqrt{T}\right)$ on DUCB, where $\sigma_G$ is a parameter characterizing the tail bound of the delays and $d$ is the feature/context dimension. For comparison, the state-of-the-art regret bound of UCB on generalized linear contextual bandits without delays is $\tilde{O}\left(d\sqrt{T}\right)$ (Filippi et al. (2010); Li et al. (2017)). Regret bounds for more general delays are also given. Note that our analysis here does not assume the number of actions to be finite, and hence these regret bounds apply to infinite-action setting as well.

Finally, we consider the important special case of linear contextual bandits with finitely many actions. In this setting, we provide a different UCB-based algorithm that estimates the underlying parameters using a biased estimator (as opposed to the unbiased estimator employed in the generalized linear contextual bandits setting) and provide a more refined analysis that achieves regret bounds which are a factor of $O(\sqrt{d})$ tighter. More specifically in this setting, as a direct comparison, when the delays are again **iid** with mean $\mu_D$, we establish a high-probability regret bound[2] of $\tilde{O}\left((1 + \mu_D + \sigma_G)\sqrt{dT}\right)$.

To the best of our knowledge, these regret bounds provide the first theoretical characterizations in (generalized) linear contextual bandits with large delays and contribute to the broad landscape of contextual bandits literature by delineating the impact of delays on performance.

## 2 Problem Setup

In this section, we describe the formulation for learning in generalized linear contextual bandits (GLCB) in the presence of delays. We start by reviewing the basics of generalized linear contextual bandits, followed by a description of the delay model. Before proceeding, we first fix some notation.

For a vector $x \in \mathbb{R}^d$, we use $\|x\|$ to denote its $l_2$-norm and $x'$ its transpose. $\mathbb{B}^d := \{x \in \mathbb{R}^d : \|x\| \leq 1\}$ is the unit ball centered at the origin. The weighted $l_2$-norm associated with a positive-definite matrix $A$ is defined by $\|x\|_A := \sqrt{x'Ax}$. The minimum and maximum singular values of a matrix $A$ are written as $\lambda_{\min}(A)$ and $\|A\|$ respectively. For two symmetric matrices $A$ and $B$ the same dimensions, $A \succeq B$ means that A-B is positive semi-definite. For a real-valued function f, we use $\dot{f}$ and $\ddot{f}$ to denote its first and second derivatives. Finally, $[n] := \{1, 2, \cdots, n\}$.

### 2.1 Generalized Linear Contextual Bandits

**Decision procedure.** We consider the generalized linear contextual bandits problem with $K$ arms. At each round $t$, the agent observes a context consisting of a set of $K$ feature vectors $x_t := \{x_{t,a} \in \mathbb{R}^d | a \in [K]\}$, which is drawn **iid** from an unknown distribution $\gamma$ with $\|x_{t,a}\| \leq 1$. Each feature vector $x_{t,a}$ is associated with an unknown stochastic reward $y_{t,a} \in [0, 1]$. If the agent selects one action $a_t$, there is a reward $y_{t,a_t} \in [0, 1]$ associated with the selected arm $a_t$ and the associated $x_{t,a_t}$. Under the classic setting, the reward is immediately observed after the decision and the information can be utilized to make decision in the next round.

**Relationship between reward $Y$ and context $X$.** In terms of the relationship between $y_{t,a_t}$ and $x_{t,a_t}$ $(t \geq 1)$, we follow the standard generalized linear contextual bandits literature (Filippi et al.

(2010); Li et al. (2017)). Define $\mathcal{H}_t^0 = \{(s, x_s, a_s, y_{s,a_s}), s \leq t-1\} \cup \{x_t\}$ as the information at the beginning of round $t$. The agent maximizes the cumulative expected rewards over $T$ rounds with information $\mathcal{H}_t^0$ at each round $t$ ($t \geq 1$). Suppose the agent takes action $a_t$ at round $t$. Denote by $X_t = x_{t,a_t}$, $Y_t = y_{t,a_t}$ and we assume the conditional distribution of $Y_t$ given $X_t$ is from the exponential family. Therefore its density is given by

$$\mathbb{P}_{\theta^*}(Y_t|X_t) = \exp\left(\frac{Y_t X_t'\theta^* - m(X_t'\theta^*)}{h(\eta)} + A(Y_t, \eta)\right). \tag{1}$$

Here, $\theta^*$ is an unknown number under the frequentist setting; $\eta \in \mathbb{R}^+$ is a given parameter; $A$, $m$ and $h$ are three normalization functions mapping from $\mathbb{R}$ to $\mathbb{R}$.

For exponential families, $m$ is infinitely differentiable, $\dot{m}(X'\theta^*) = \mathbb{E}[Y|X]$, and $\ddot{m}(X'\theta^*) = \mathbb{V}(Y|X)$. Denote $g(X'\theta^*) = \mathbb{E}[Y|X]$, one can easily verify that $g(x'\theta) = x'\theta$ for linear model, $g(x'\theta) = \frac{1}{1+\exp(-x'\theta)}$ for logistic model and $g(x'\theta) = \exp(x'\theta)$ for Poisson model. In the generalized linear model (GLM) literature (Nelder and Wedderburn (1972); McCullagh (2018)), $g$ is often referred to as the *inverse link function*. Note that (1) can be rewritten as the GLCB form,

$$Y_t = g(X_t'\theta^*) + \epsilon_t, \tag{2}$$

where $\{\epsilon_t, t \in [T]\}$ are independent zero-mean noise, $\mathcal{H}_t^0$-measurable with $\mathbb{E}[\epsilon_t|\mathcal{H}_t^0] = 0$. Data generated from (1) automatically satisfies the sub-Gaussian condition:

$$\mathbb{E}\left[\exp(\lambda\epsilon_t)|\mathcal{H}_t^0\right] \leq \exp\left(\frac{\lambda^2\sigma^2}{2}\right). \tag{3}$$

Throughout the paper, we denote $\sigma > 0$ as the sub-Gaussian parameter of the noise $\epsilon_t$.

**Remark 1.** *In this paper, we focus on the GLM with exponential family (1). In general, one can work with model (2) under the sub-Gaussian assumption (3). Our analysis will still hold by considering maximum quasi-likelihood estimator for (2). See more explanations in the appendix.*

## 2.2 The Delay Model

Unlike the traditional setting where each reward is immediately observed, here we consider the case where stochastic and unbounded *delays* are present in revealing the rewards. Let $T$ be the number of total rounds. At round $t$, after the agent takes action $a_t$, the reward $y_{t,a_t}$ may not be available immediately. Instead, it will be observed at the end of round $t + D_t$ where $D_t$ is the delay at time $t$. We assume $D_t$ is a non-negative random number which is independent of $\{D_s\}_{s \leq t-1}$ and $\{x_s, y_{s,a_s}, a_s\}_{s \leq t}$. First, we define the available information for the agent at each round.

**Information structure under delays.** At any round $t$, if $D_s + s \leq t - 1$ (reward occurred in round $s$ is available at the beginning of round $t$), then we call $(s, x_s, y_{s,a_s}, a_s)$ the *complete information tuple* at round $t$. If $D_s + s \geq t$, we call $(s, x_s, a_s)$ the *incomplete information tuple* at the beginning of round $t$. Define

$$\mathcal{H}_t = \{(s, x_s, y_{s,a_s}, a_s) \mid s + D_s \leq t - 1\} \cup \{(s, x_s, a_s) \mid s \leq t-1, s + D_s \geq t\} \cup \{x_t\},$$

then $\mathcal{H}_t$ is the information (filtration) available at the beginning of round $t$ for the agent to choose action $a_t$. In other words, $\mathcal{H}_t$ contains all the incomplete and complete information tuples up to round $t - 1$ and the content vector $x_t$ at round $t$.

Moreover define

$$\mathcal{F}_t = \{(s, x_s, a_s, y_{s,a_s}) \mid s + D_s \leq t\}. \tag{4}$$

Then $\mathcal{F}_t$ contains all the complete information tuples $(s, x_s, a_s, y_{s,a_s})$ up to the end of round $t$. Denote $\mathcal{I}_t = \mathcal{F}_t - \mathcal{F}_{t-1}$, $\mathcal{I}_t$ is the new complete information tuples revealed at the end of round $t$.

**Performance criterion.** Under the frequentist setting, assume there exists an unknown true parameter $\theta^* \in \mathbb{R}^d$. The agent's strategy can be evaluated by comparing her rewards to the best reward. To do so, define the optimal action at round $t$ by $a_t^* = \arg\max_{a \in [K]} g(x_{t,a}'\theta^*)$. Then, the agent's total regret of following strategy $\pi$ can be expressed as follows

$$R_T(\pi) := \sum_{t=1}^{T}\left(g\left(x_{t,a_t^*}'\theta^*\right) - g\left(x_{t,a_t}'\theta^*\right)\right),$$

where $a_t \sim \pi_t$ and policy $\pi_t$ maps $\mathcal{H}_t$ to the probability simplex $\Delta^K := \{(p_1, \cdots, p_K) \mid \sum_{i=1}^{K} p_i = 1, p_i \geq 0\}$. Note that $R_T(\pi)$ is in general a random variable due to the possible randomness in $\pi$.

**Assumptions.** Through out the paper, we assume the following assumption on distribution $\gamma$ and function $g$, which is standard in the generalized linear bandit literature (Filippi et al. (2010); Li et al. (2017); Jun et al. (2017)).

**Assumption 1** (GLCB). • $\lambda_{\min}(\mathbb{E}[\frac{1}{K}\sum_{a\in[K]} x_{t,a}x'_{t,a}]) \geq \sigma_0^2$ *for all $t \in [T]$.*

• $\kappa := \inf_{\{\|x\|\leq 1, \|\theta-\theta^*\|\leq 1\}} \dot{g}(x'\theta) > 0.$

• *g is twice differentiable. $\dot{g}$ and $\ddot{g}$ are upper bounded by $L_g$ and $M_g$, respectively.*

In addition, we assume the delay sequence $\{D_t\}_{t=1}^T$ satisfies the following assumption.

**Assumption 2** (Delay). *Assume $\{D_t\}_{t=1}^T$ are independent non-negative random variables with tail-envelope distribution $(\xi_D, \mu_D, M_D)$. That is, there exists a constant $M_D > 0$ and a distribution $\xi_D$ with mean $\mu_D < \infty$ such that for any $m \geq M_D$ and $t \in [T]$,*

$$\mathbb{P}(D_t \geq m) \leq \mathbb{P}(D \geq m),$$

*where $D \sim \xi_D$ and $\mathbb{E}[D] = \mu_D$. Furthermore, assume there exists $q > 0$ such that*

$$\mathbb{P}(D - \mu_D \geq x) \leq \exp\left(\frac{-x^{1+q}}{2\sigma_D^2}\right).$$

Note that when $q = 1$, $D$ is sub-Gaussian with parameter $\sigma_D$. When $q \in (0, 1)$, $D$ has near-heavy tail distribution. When $D_i$'s are **iid**, the following condition guarantees Assumption 2:

$$\mathbb{P}(D_i - \mathbb{E}[D_i] \geq x) \leq \left(\frac{-x^{1+q}}{2\tilde{\sigma}_D^2}\right),$$

with some $\tilde{\sigma}_D > 0$ and $q > 0$.

For ease of reference (as there are many floating parameters in this paper), we summarize all the parameter definitions in Table 1.

| Notation | Definition | Notation | Definition |
|---|---|---|---|
| $K$ | number of arms | $\xi_D$ | tail-envelope distribution for the delays |
| $d$ | feature dimension | $q$ | parameter of $\xi_D$ |
| $\kappa$ | $\inf_{\{\|x\|\leq 1, \|\theta-\theta^*\|\leq 1\}} \dot{g}(x'\theta)$ | $\mu_D$ | expectation of $\xi_D$ |
| $\theta^*$ | unknown true parameter | $M_D$ | parameter of $\xi_D$ |
| $\sigma$ | sub-Gaussian parameter for $\epsilon_t$ | $\sigma_D$ | parameter of $\xi_D$ |
| $L_g$ | upper bound on $\dot{g}$ | $\sigma_G$ | sub-Gaussian parameter of $G_t$ |
| $M_g$ | upper bound on $\ddot{g}$ | $\mu'_D$ | expectation of **iid** delays |
| $\sigma_0^2$ | lower bound on $\lambda_{\min}(\mathbb{E}[\frac{1}{K}\sum_{a\in[K]} x_{t,a}x'_{t,a}])$ | $D_{max}$ | upper bound on bounded delays |

Table 1: Parameters in the GLCB model with delays.

## 3 Delayed Upper Confidence Bound (DUCB) for GLCB

In this section, we propose a UCB type of algorithm for GLCB, adapting the delay information in an online version. Let us first define some variables and state the main algorithm.

### 3.1 Algorithm: DUCB-GLCB

Denote $G_t = \sum_{s=1}^{t-1} \mathbb{I}\{s + D_s \geq t\}$ as the number of missing reward when the agent is making a prediction at round $t$. Denote $T_t = \{s : s \leq t - 1, D_s + s \leq t - 1\}$ as the set containing timestamps with complete information tuples at the beginning of round $t$. Further denote $W_t = \sum_{s\in T_t} X_s X'_s$ as the matrix consisting feature information with timestamps in $T_t$ and $V_t = \sum_{s=1}^{t-1} X_s X'_s$ as the matrix consisting all available features at the end of round $t - 1$. The main algorithm is given below.

---
**Algorithm 1 DUCB-GLCB**

---
1: **Input**: the total rounds $T$, model parameters $d$ and $\kappa$, and tuning parameters $\tau$ and $\delta$.
2: **Initialization**: randomly choose $\alpha_t \in [K]$ for $t \in [\tau]$, set $V_{\tau+1} = \sum_{i=1}^{\tau} X_s X_s'$, $T_{\tau+1} := \{s : s \le \tau, s + D_s \le \tau\}$, $G_{\tau+1} = \tau - |T_{\tau+1}|$ and $W_{\tau+1} = \sum_{s \in T_{\tau+1}} X_s X_s'$
3: **for** $t = \tau+1, \tau+2, \cdots, T$ **do**
4:     **Update Statistics**: calculate the MLE $\hat{\theta}_t$ by solving $\sum_{s \in T_t}(Y_s - g(X_s'\theta))X_s = 0$
5:     **Update Parameter**: $\beta_t = \frac{\sigma}{\kappa}\sqrt{\frac{d}{2}\log\left(1 + \frac{2(t - G_t)}{d}\right) + \log(\frac{1}{\delta})} + \sqrt{G_t}$
6:     **Select Action**: choose $a_t = \arg\max_{a \in [K]}\left(x_{t,a}'\hat{\theta}_t + \beta_t\|x_{t,a}\|_{V_t^{-1}}\right)$
7:     **Update Observations**: $X_t \leftarrow x_{t,a_t}$, $V_{t+1} \leftarrow V_t + X_t X_t'$ and $T_{t+1} \leftarrow T_t \cup \{s : s + D_s = t\}$, $G_{t+1} = t - |T_{t+1}|$, and $W_{\tau+1} = W_\tau + \sum_{s:s+D_s = t} X_s X_s'$
8: **end for**

---

**Remark 2.** *In step 4, we use Maximum Likelihood Estimators (MLEs) for the parameter estimation step at each round $t$. For more details on the derivation and explanation, we refer to the appendix.*

**Remark 3** (Comparison to UCB-GLM Algorithm in Li et al. (2017))**.** *We make several adjustments to the UCB-GLM Algorithm in Li et al. (2017). First, in step 4 (statistics update), we only use data with timestamps in $T_t$ to calculate the estimator using MLE. In this step, using data without reward will cause bias in the estimation. Second, when selecting the action in step 5, parameter $\beta_t$ is updated adaptively at each round whereas in Li et al. (2017), the corresponding parameter is constant over time. Moreover, in step 4, we choose to use $V_t$ to normalize the context vector $X_{t,a}$ instead of $W_t$.*

## 3.2 Preliminary Analysis

Denote $G_t^* = \max_{1 \le s \le t} G_s$ as the running maximum number of missing reward up to round $t$. The property of $G_t$ and $G_t^*$ is the key to analyze the regret bound for UCB algorithm. We next characterize the tail behavior of $G_t$ and $G_t^*$.

**Proposition 1** (Properties of $G_t$ and $G_t^\star$)**.** *Assume Assumption 2. Denote $\sigma_G = \sqrt{\frac{I}{4} + \frac{\sigma_D^2(1+q)}{q}}$ with $I = \max\left\{\sqrt[1+q]{2\log(2)\sigma_D^2}, \sqrt[q]{\frac{2\sigma_D^2}{1+q}} + 1\right\}$. Then,*

1. *$G_t$ is sub-Gaussian. Moreover, for all $t \ge 1$,*

$$\mathbb{P}\left(G_t \ge 2(\mu_D + M_D) + x\right) \le \exp\left(\frac{-x^2}{2\sigma_G^2}\right). \tag{5}$$

2. *With probability $1 - \delta$,*

$$G_T^* \le 2(\mu_D + M_D) + \sigma_G\sqrt{2\log(T)} + \sigma_G\sqrt{2\log\left(\frac{1}{\delta}\right)}, \tag{6}$$

    *where $G_T^* = \max_{1 \le s \le T} G_s$.*

3. *Define $W_t = \sum_{s \in T_t} X_s X_s'$ where $X_t$ is drawn **iid**. from some distribution $\gamma$ with support in the unit ball $\mathbb{B}_d$. Furthermore, let $\Sigma := \mathbb{E}[X_t X_t']$ be the second moment matrix, and $B$ and $\delta > 0$ be two positive constants. Then there exist positive, universal constants $C_1$ and $C_2$ such that $\lambda_{\min}(W_t) \ge B$ with probability at least $1 - 2\delta$, as long as*

$$t \ge \left(\frac{C_1\sqrt{d} + C_2\sqrt{\log(\frac{1}{\delta})}}{\lambda_{\min}(\Sigma)}\right)^2 + \frac{2B}{\lambda_{\min}(\Sigma)} + 2(\mu_D + M_D) + \sigma_G\sqrt{2\log\left(\frac{1}{\delta}\right)}. \tag{7}$$

The proof of Proposition 1 is deferred to the appendix. This Note that $G_t$ is sub-Gaussian even when $D$ has near-heavy tail distribution when $p \in (0,1)$.

## 3.3 Regret Bounds

**Theorem 2.** *Assume Assumptions 1-2. Fix any $\delta$. There exists a universal constant $C := C(C_1, C_2, M_D, \mu_D, \sigma_0, \sigma_G, \sigma, \kappa) > 0$, such that if we run DUCB-GLCB with $\tau := C\left(d + \log(\frac{1}{\delta})\right)$ and $\beta_t = \frac{\sigma}{\kappa}\sqrt{\frac{d}{2}\log\left(1 + \frac{2(t-G_t)}{d}\right) + \log(\frac{1}{\delta})} + \sqrt{G_t}$, then, with probability at least $1 - 5\delta$, the regret of the algorithm is upper bounded by*

$$
\begin{aligned}
R_T \quad \leq \quad & \tau + L_g \left[ 4\sqrt{\mu_D + M_D}\sqrt{Td\log\left(\frac{T}{d}\right)} + 2^{7/4}\sqrt{\sigma_G}\left(\log\left(\frac{1}{\delta}\right)\right)^{1/4}\sqrt{d\log\left(\frac{T}{d}\right)T} \right. \\
& \left. + 2^{7/4}\sqrt{\sigma_G}\left(\log\left(T\right)\right)^{1/4}\sqrt{d\log\left(\frac{T}{d}\right)T} + \frac{2d\sigma}{\kappa}\log\left(\frac{T}{d\delta}\right)\sqrt{T} \right].
\end{aligned}
\tag{8}
$$

For parameter definition, we refer to Table 1.The proof of Theorem 2 is deferred to the appendix.

**Corollary 3** (Expected regret). *Assume Assumptions 1-2. The expected regret is bounded by*

$$
\mathbb{E}[R_T] = O\left(d\sqrt{T}\log(T) + \sqrt{\mu_D + M_D}\sqrt{Td\log(T)} + \sqrt{\sigma_G}\sqrt{Td}\left(\log(T)\right)^{3/4}\right).
\tag{9}
$$

Given the result in (8), (9) holds by choosing $\delta = \frac{1}{T}$ and using the fact that $R_T \leq T$.

The highest order term $O(d\sqrt{T}\log(T))$ does not depend on delays. Delay impacts the expected regret bound in two folds. First, the sub-Gaussian parameter $\sigma_G$ appears in the second-highest order term. Second, the mean-related parameter $\mu_D + M_D$ appears in the third-order term. Note that here we include the log factors in deciding the highest order term, the second higest order term and so on. If we exclude the log terms, then both delay parameters impact the regret bound multiplicatively.

## 3.4 Tighter Regret Bounds for Special Cases

When the sequence $\{D_s\}_{s=1}^T$ satisfies some specific assumptions, we are able to provide tighter high probability bounds on the regret.

**Proposition 4.** *Under Assumption 1, we have:*

1. *If there exists a constant $D_{\max} > 0$ such that $\mathbb{P}(D_s \leq D_{\max}) = 1$ for all $s \in [T]$. Fix $\delta$. There exists a universal constant $C > 0$ such that by taking $\tau = D_{\max} + C(d + \log(\frac{1}{\delta}))$, with probability $1 - 3\delta$, the regret of the algorithm is upper bounded by*

$$
R_T \leq \tau + L_g\left(2\sqrt{D_{\max}}\sqrt{2Td\log\left(\frac{T}{d}\right)} + \frac{2d\sigma}{\kappa}\log\left(\frac{T}{d\delta}\right)\sqrt{T}\right).
\tag{10}
$$

   *Therefore, $\mathbb{E}[R_T] = O\left(\sqrt{D_{\max}}\sqrt{dT\log(T)} + d\sqrt{T}\log(T)\right)$.*

2. *Assume $D_1, \cdots, D_T$ are **iid** non-negative random variables with mean $\mu_D'$ that satisfy Assumption (2). There exists $C > 0$ such that by taking $\tau := C\left(d + \log(\frac{1}{\delta})\right)$, with probability $1 - 5\delta$, the regret of the algorithm is upper bounded by*

$$
\begin{aligned}
R_T \quad \leq \quad & \tau + L_g \left[ 4\sqrt{\mu_D'}\sqrt{Td\log\left(\frac{T}{d}\right)} + 2^{7/4}\sqrt{\sigma_G}\left(\log\left(\frac{1}{\delta}\right)\right)^{1/4}\sqrt{d\log\left(\frac{T}{d}\right)T} \right. \\
& \left. + 2^{7/4}\sqrt{\sigma_G}\left(\log\left(T\right)\right)^{1/4}\sqrt{d\log\left(\frac{T}{d}\right)T} + \frac{2d\sigma}{\kappa}\log\left(\frac{T}{d\delta}\right)\sqrt{T} \right].
\end{aligned}
$$

   *Therefore, $\mathbb{E}[R_T] = O\left(\left(\sqrt{\mu_D'} + \sqrt{\sigma_G}\log(T)^{3/4}\right)\sqrt{Td} + d\log(T)\sqrt{T}\right)$*

# 4 Tighter Regret Bounds on Linear Contextual Bandits with Finite Actions

We now consider the important special case of linear contextual bandits. and tighten the $O(d)$ dependence from previous bounds to $O(\sqrt{d})$. This requires two new elements that we incorporate into DUCB-GLCB in Algorithm 1. First, instead of using MLE which is unbiased, here we use an unbiased estimator that incorporates all the contexts (including those contexts for which the rewards have not been received). In the linear contextual bandits setting, one can obtain analytical formulas for the estimation procedure. Second, we extend the Sup-Base UCB decomposition framework (first devised in Auer (2002) and subsequently adapted in Chu et al. (2011); Li et al. (2017)) to the current setting in order to resolve the reward dependency issue. This framework is a commonly used one in the literature that deals with the dependency issue, and provides a $O(\sqrt{dT})$ regret bound instead of a $O(d\sqrt{T})$ regret bound. Here we adapt this framework in the delayed reward setting.

In summary, the algorithm has two components, Delayed BaseLinUCB (Algorithm 2) and Delayed SupLinUCB (Algorithm 3). Delayed BaseLinUCB performs estimation and the confidence bound computation, using a subset $\Psi_t$ of the past time steps as opposed to the set of all past time steps (note that when $t = 1$, the chosen subset $\Psi_t$ is necessarily the empty set). This subset is carefully chosen in Delayed SupLinUCB to make sure rewards are indepenent when conditioned on the past selected contexts. Delayed SupLinUCB is further responsible for selecting an action at each time step.

---

**Algorithm 2 Delayed BaseLinUCB at Step t**

---

1: **Input**: $\Psi_t \subset \{1, 2, \cdots, t-1\}$.
2: $A_t = I_d + \sum_{\tau \in \Psi_t} x_{t,a_\tau} x'_{t,a_\tau}$
3: $c_t = \sum_{\tau \in \Psi_t} \mathbf{1}(D_\tau + \tau \leq t - 1) y_{\tau,a_\tau} x_{\tau,a_\tau}$
4: $\theta_t = A_t^{-1} c_t$
5: Observe $K$ arm features, $x_{t,1}, x_{t,2}, \cdots, x_{t,K} \in \mathbb{R}^d$
6: **for** $a \in [K]$ **do**
7: $\quad w_{t,a} = \alpha_t \sqrt{x_{t,a}^T A_t^{-1} x_{t,a}}$
8: $\quad \hat{y}_{t,a} \leftarrow \theta_t^T x_{t,a}$
9: **end for**

---

---

**Algorithm 3 Delayed SupLinUCB**

---

1: **Input**: $T \in \mathbb{N}$, $S \leftarrow \log(T)$
2: $\Psi_1^s \leftarrow \emptyset$ for all $s \in [T]$
3: **for** $t = 1, 2, \cdots, T$ **do**
4: $\quad s \leftarrow 1$ and $\hat{A}_1 \leftarrow [K]$
5: $\quad$ **repeat**
6: $\quad\quad$ Use Delayed BaseLinUCB with $\Psi_t^s$ to calculate the width, $w_{t,a}^s$, and upper confidence bound, $\hat{y}_{t,a}^s + w_{t,a}^s$, for all $a \in \hat{A}_s$
7: $\quad\quad$ **if** $w_{t,a}^s \leq 1/\sqrt{T}$ for all $a \in \hat{A}_s$ **then**
8: $\quad\quad\quad$ Choose $a_t = \arg\max_{a \in \hat{A}_s} \left(\hat{y}_{t,a}^s + w_{t,a}^s\right)$, Update $\Psi_{t+1}^{s'} \leftarrow \Psi_t^{s'}$ for all $s' \in [S]$.
9: $\quad\quad$ **else if** $w_{t,a}^s \leq 2^{-s}$ for all $a \in \hat{A}_s$ **then**
10: $\quad\quad\quad$ $\hat{A}_{s+1} \leftarrow \{a \in \hat{A}_s \mid \hat{y}_{t,a}^s + w_{t,a}^s \geq \max_{a' \in \hat{A}_s}(\hat{y}_{t,a'}^s + w_{t,a'}^s) - 2^{1-s}\}$, $s \leftarrow s + 1$
11: $\quad\quad$ **else**
12: $\quad\quad\quad$ Choose $a_t \in \hat{A}_s$ such that $w_{t,a_t}^s > 2^{-s}$, Update

$$\Phi_{t+1}^{s'} \leftarrow \begin{cases} \Phi_{t,}^{s'} \cup \{t\} & \text{if } s = s' \\ \Phi_{t,}^{s'} & \text{otherwise} \end{cases}$$

13: $\quad\quad$ **end if**
14: $\quad$ **until** an action $a_t$ is found.
15: **end for**

---

**Remark 4.** *There are two modifications compared to Algorithm 2 in Chu et al. (2011). First, the estimator $\theta_t$ (in step 4) is a biased estimator. We use all the features in matrix $A_t$ and only use features with observed rewards in vector $c_t$. In particular, when the indicator $\mathbf{1}(D_\tau + \tau \leq t-1)$ evaluates to 1, the reward corresponding to the action taken at time step $\tau$ has been received by the end of (and possibly prior to) $t-1$ (and hence available at the beginning of $t$); all the other rewards (i.e. those that have not been received by $t-1$) are excluded. In comparison, Chu et al. (2011) construct an unbiased estimator in each time step. Second, the width parameter $\alpha_t$ (in step 7) is time-dependent and adapts to new information (based on the delays) in each round. In comparison, the width parameter is constant in Chu et al. (2011) that only depends on the horizon $T$.*

**Theorem 5** (Regret on Delayed SupLinUCB-BaseLinUCB). *If Delayed SupLinUCB is run with $\alpha_t = \bar{\alpha} + G_t + 1$, where $\bar{\alpha} = \sqrt{\frac{1}{2}\ln\left(\frac{2TK\log(T)}{\delta}\right)}$, then with probability at least $1 - 2\delta$, the regret of the algorithm is*

$$O\left(\sqrt{Td}\left((\sigma_G + 1)\log^{3/2}(\frac{TK\log T}{\delta}) + \log(\frac{TK\log T}{\delta})(1 + \mu_D + M_D + \sigma_G\sqrt{\log\frac{1}{\delta}})\right)\right). \quad (11)$$

The proof of Theorem 5 requires of modification of two lemmas in Chu et al. (2011). Lemma 6 is a modification of (Chu et al., 2011, Lemma 1) and Lemma 7 is a modification of (Chu et al., 2011, Lemma 6). We defer the detailed proofs of Lemmas 6-7 to the appendix. Proof of Theorem 5 is also given in the appendix.

In the regret bound (11), the delay parameters $(\mu_D, M_D, \sigma_D)$ appear on the highest order term $\sqrt{Td}$. Although the highest order term $\sqrt{Td}$ is removed from (8), the delay on order $O(\sqrt{Td})$ is essential and this is also true for (8).

**Lemma 6.** *Suppose the input index set $\Phi_t$ in Delayed BaseLinUCB is constructed so that for fixed $x_{\tau,a_\tau}$ with $\tau \in \Phi_t$, the rewards $y_{\tau,a_\tau}$ are independent random variables with means $\mathbb{E}[y_{\tau,a_\tau}] = x'_{\tau,a_\tau}\theta^*$. Suppose $\{G_t\}$ is fixed and given. Then, with probability at least $1 - \delta/T$, we have for all $a \in [K]$ that*

$$|\hat{y}_{t,a} - x'_{t,a}\theta^*| \leq \left(1 + \sqrt{\frac{1}{2}\ln\left(\frac{2TK}{\delta}\right)} + G_t\right)s_{t,a}.$$

**Lemma 7.** *Assume $G_T^*$ is fixed and given. For all $s \in [S]$,*

$$\Psi_{T+1}^s \leq 5 \cdot 2^s \left(\sqrt{2}\bar{\alpha}(G_T^* + \bar{\alpha})\right)\sqrt{d|\Psi_{T+1}^s|}$$

**Remark 5** (Why Assumption 1 can be dropped in Theorem 5). *There are essentially two methods to guarantee a positive lower bound $\lambda_{\min}(\sum_{s=1}^t X_s X_s')$. One method is to randomly sample actions for $\tau$ rounds. In this way, (Li et al., 2017, Proposition 1) guarantees a positive lower bound on $\lambda_{\min}(\sum_{s=1}^t X_s X_s')$. This is the method adopted in Algorithm 1 and Theorem 2. The other method adds a regularization term. This is adopted in the definition of $A_t$ (See Algorithm 2 and Theorem 5). This method corresponds to the Ridge regression when estimating parameter $\theta_t$.*

## 5 Conclusion

Beyond contextual bandits and looking at the broader landscape of data-driven decision making, delays have emerged to be an important phenomenon in several domains, including, among other things, distributed stochastic optimization (Bertsekas and Tsitsiklis (1997); Zhou et al. (2018b)), multi-agent game-theoretical and reinforcement learning (Zhou et al. (2017); Grover et al. (2018a); Guo et al. (2019); Mertikopoulos and Zhou (2019)), real-time scheduling in large-scale systems (Pinedo; Mehdian et al. (2017); Mahdian et al. (2018)). Data-driven decision making with imperfect information is an emerging research paradigm and much remains to be understood in regards to how decision-making needs to be adapted in the presence of delays.

## Footnotes

*These two authors contributed equally.

[2]In this case, the number of actions being finite is important. In particular, the regret bound has a $O(\log K)$ dependence. Consequently, strictly speaking, if $K$ is not viewed as a constant, we would also need $K$ to not be too large compared to $d$ in order to retain the same regret bound of $\tilde{O}\left((\mu_D + \sigma_G + 1)\sqrt{dT}\right)$. A common (and rather weak) assumption is the $K$ is polynomial in $d$.

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
