[Supplementary Material · delay_camera_ready_appendix_v5.pdf]

# Supplementary for

## Learning in Generalized
## Linear Contextual Bandits with Stochastic Delays

## A    Table of Parameters

| Notation | Definition |
|---:|---|
| $K$ | number of arms |
| $d$ | feature dimension |
| $\kappa$ | $\inf_{\{\|x\|\leq 1, \|\theta - \theta^*\| \leq 1\}} \dot{g}(x'\theta)$ |
| $\theta^*$ | unknown parameter in GLCB model |
| $\sigma$ | sub-Gaussian parameter for noise $\epsilon_t$ |
| $L_g$ | upper bound on $\dot{g}$ |
| $M_g$ | upper bound on $\ddot{g}$ |
| $\sigma_0^2$ | lower bound on $\lambda_{\min}(\mathbb{E}[\frac{1}{K}\sum_{a\in[K]} x_{t,a}x'_{t,a}])$ |
| $\xi_D$ | tail-envelope distribution for the delays |
| $q$ | parameter to characterize the tail-envelope distribution $\xi_D$ |
| $\mu_D$ | expectation of the tail-envelope distribution $\xi_D$ |
| $M_D$ | parameter of $\xi_D$ |
| $\sigma_D$ | parameter of $\xi_D$ |
| $\sigma_G$ | sub-Gaussian parameter of $G_t$ |
| $\mu'_D$ | expectation of **iid** delays |
| $D_{max}$ | upper bound on bounded delays |

Table 2: Parameters in the GLCB model with delays.

## B    Auxiliary Results

**Theorem 8** (Maximum over a finite set, Wainwright (2019))**.** *Let $X_1, \cdots, X_n$ be centered $\sigma$-sub-Gaussian random variables. (i.e. $\mathbb{E}[\exp(\lambda X_i)] \leq \exp\left(\frac{\lambda^2\sigma^2}{2}\right)$). Then,*

$$\mathbb{E}\left(\max_{1\leq i\leq n} X_i\right) \leq \sigma\sqrt{2\log(n)},$$

*and*

$$\mathbb{E}\left(\max_{1\leq i\leq n} |X_i|\right) \leq \sigma\sqrt{2\log(2n)}.$$

*Moreover, for any $t \geq 0$,*

$$\mathbb{P}(\max_{1\leq i\leq n} X_i > t) \leq \exp\left(-\frac{t^2}{2\sigma^2} + \log n\right),$$

*and*

$$\mathbb{P}(\max_{1\leq i\leq n} |X_i| > t) \leq 2\exp\left(-\frac{t^2}{2\sigma^2} + \log n\right).$$

Note that the random variables in Theorem 8 need not be independent.

**Theorem 9** ( Sub-Gaussian parameter for centered indicator random variables, Ostrovsky and Sirota (2014) ). *Let $p \in [0,1]$ and let $\eta$ be a centered random variable such that $\mathbb{P}(\eta = 1 - p) = p$ and $\mathbb{P}(\eta = -p) = 1 - p$, then*

$$\mathbb{E}[\exp(\lambda\eta)] \leq \exp(\lambda^2 Q(p)),$$

*where $Q(p) = \frac{1-2p}{4\log(\frac{1-p}{p})}$.*

**Theorem 10** (Hoeffding Bound, Wainwright (2019)). *Let $X_1, \cdots, X_n$ be independent random variables. Assume $X_i$ has mean $\mu_i$ and sub-Gaussian parameter $\sigma_i$. Then for all $t \geq 0$, we have*

$$\mathbb{P}\left(\sum_{i=1}^{n}(X_i - \mu_i) \geq t\right) \leq \exp\left(-\frac{t^2}{2\sum_{i=1}^{n}\sigma_i^2}\right).$$

## C Maximum Likelihood Estimators (MLEs).

We use data with timestamps in $T_t$ to construct the MLE. Suppose we have independent samples of $\{Y_s : s \in T_t\}$ condition on $\{X_s : s \in T_t\}$. The log-likelihood function of $\theta$ under (1) is

$$
\begin{aligned}
\log l\left(\theta \mid T_t\right) &= \sum_{s \in T_t}\left[\frac{Y_s X_s'\theta - m(X_s'\theta)}{v(\eta)} + B(Y_s, \eta)\right] \\
&= \frac{1}{v(\eta)}\sum_{s \in T_t}\left[Y_s X_s'\theta - m(X_s'\theta)\right] + \text{constant}.
\end{aligned}
$$

Therefore, the MLE can be defined as

$$\hat{\theta}_t \in \arg\max_{\theta \in \Theta}\sum_{s \in T_t}\left[Y_s X_s'\theta - m(X_s'\theta)\right].$$

Since $m$ is differentiable with $\ddot{m} \geq 0$, the MLE can be written as the solution of the following equation

$$\sum_{s \in T_t}(Y_s - g(X_s'\theta))X_s = 0, \tag{12}$$

which is the estimator we use in Step 4 of Algorithm 1.

Note that, the general GLCB, a semi-parametric version of the GLM, is obtained by assuming only that $\mathbb{E}[Y|X] = g(X'\theta^*)$ (see (2)) without further assumptions on the conditional distribution of $Y$ given $X$. In this case, the estimator obtained by solving (12) is referred to as the *maximum quasi-likelihood estimator*. It is well-documented that this estimator is consistent under very general assumptions as long as matrix $\sum_{s \in T_t}X_s X_s'$ tends to infinity as $t \to \infty$ (Chen et al. (1999); Filippi et al. (2010)).

## D Missing Proofs

In this section, we provide the proofs of Propostion 1, Theorem 2, Proposition 4, Lemma 6, Lemma 7 and Theorem 5.

*Proof of Proposition 1.* Now let us prove the three properties in Proposition 1.

**Property 1.** Let $\tilde{D}_{k_i}$ be a random variable such that $\tilde{D}_{k_i} \geq -(\mu_D + M_D)$ almost surely, $\mathbb{E}[\tilde{D}_{k_i}] \leq 0$ and $\mathbb{P}(\tilde{D}_{k_i} \geq x) \leq \exp\left(-\frac{x^{1+q}}{2\sigma_D^2}\right)$ for $x \geq 0$. One can view $\tilde{D}_{k_i}$ as a shifted delay.

Define $\tilde{I}_i = \mathbb{I}\left(\tilde{D}_{k_i} \geq i\right) - p_i$ with $p_i = \mathbb{P}(\tilde{D}_{k_i} \geq i)$. Then $\mathbb{P}\left(\tilde{I}_i = 1 - p_i\right) = p_i$ and $\mathbb{P}(\tilde{I}_i = p_i) = 1 - p_i$. Denote $\sigma_i = \sqrt{\frac{1-2p_i}{2\log\left(\frac{1-p_i}{p_i}\right)}}$, it is easy to verify that

$$\mathbb{E}\exp\left(\lambda\tilde{I}_i\right) = p_i\exp(\lambda(1 - p_i)) + (1 - p_i)\exp(-p_i\lambda) \leq \exp\left(\frac{\sigma_i^2\lambda^2}{2}\right).$$

Therefore $\tilde{I}_i$ is sub-Gaussian with parameter $\sigma_i$. (Also see Theorem 9.)

We first show that when $i \geq \max \left\{ \sqrt[1+q]{2 \log(2)\sigma_D^2}, \sqrt[q]{\frac{2\sigma_D^2}{1+q}} + 1 \right\} := I$, the following two facts hold:

$$p_i \leq \frac{1}{2}, \tag{13}$$

$$\text{and} \quad \exp\left(\frac{i^{1+q}}{2\sigma_D^2}\right) - \exp\left(\frac{(i-1)^{1+q}}{2\sigma_D^2}\right) \geq 1. \tag{14}$$

- When $i \geq \sqrt[1+q]{2\log(2)\sigma_D^2}$,

$$p_i \leq e^{-\frac{i^{1+q}}{2\sigma_D^2}} \leq \frac{1}{2}.$$

The first inequality holds by Assumption 2 and second inequality holds by simple calculation.

- Define $h(x) = \exp\left(\frac{x^{1+q}}{2\sigma_D^2}\right)$ with $q > 0$, which is differentiable. By Mean Value Theorem, $h(x) - h(y) = \exp\left(\frac{z^{1+q}}{2\sigma_D^2}\right) \frac{(1+q)z^q}{2\sigma_D^2} (x - y)$ for some $z \in (x, y)$. Take $x = i - 1$ and $y = i$, for some $z \in (i-1, i)$, we have

$$
\begin{aligned}
\exp\left(\frac{i^{1+q}}{2\sigma_D^2}\right) - \exp\left(\frac{(i-1)^{1+q}}{2\sigma_D^2}\right) &= \exp\left(\frac{z^{1+q}}{2\sigma_D^2}\right) \frac{(1+q)z^q}{2\sigma_D^2} \\
&\geq \frac{(1+q)z^q}{2\sigma_D^2} \geq \frac{(1+q)(i-1)^q}{2\sigma_D^2} \geq 1. \quad (15)
\end{aligned}
$$

The last inequality in (15) holds since $i \geq \sqrt[q]{\frac{\sigma^2}{1+q}} + 1$.

Given (13)-(14), when $i \geq I$ and $q \geq 0$,

$$\sigma_i^2 = \frac{1 - 2p_i}{2 \log\left(\frac{1-p_i}{p_i}\right)} \quad \leq \quad \frac{1}{2 \log\left(\frac{1-p_i}{p_i}\right)} \tag{16}$$

$$\leq \quad \frac{\sigma_D^2}{(i-1)^{1+q}}. \tag{17}$$

(16) holds since (13) and (17) holds since (14). Therefore

$$
\begin{aligned}
\sum_{i=I}^{\infty} \sigma_i^2 &= \sum_{i=I}^{\infty} \frac{1 - 2p_i}{2 \log\left(\frac{1-p_i}{p_i}\right)} \leq \sum_{i=I}^{\infty} \frac{1}{2 \log\left(\frac{1-p_i}{p_i}\right)} \leq \sum_{i=I-1}^{\infty} \frac{\sigma_D^2}{i^{1+q}} \\
&\leq \sigma_D^2 \left(1 + \sum_{i=2}^{\infty} \frac{1}{i^{1+q}}\right) \leq \sigma_D^2 \left(1 + \int_1^{\infty} \frac{1}{x^{(1+q)}} dx\right) = \frac{\sigma_D^2(1+q)}{q}.
\end{aligned}
$$

It is easy to check that $\sigma_i^2 = \frac{1-2p_i}{2\log\left(\frac{1-p_i}{p_i}\right)} \leq \frac{1}{4}$ for all $p_i \in [0,1]$. Therefore, $\sum_{i=1}^{\infty} \sigma_i^2 \leq \frac{1}{4}I + \frac{\sigma_D^2(1+q)}{q}$.

Define $\tilde{G} = \sum_{i=1}^{\infty} \tilde{I}_i$. combining above result with Theorem 10, $\tilde{G}$ is sub-Gaussian with parameter $\sigma_G = \sqrt{\frac{I}{4} + \frac{\sigma_D^2(1+q)}{q}}$. Similarly, we can show that $\tilde{G}_t = \sum_{i=1}^{t} \tilde{I}_i$ is sub-Gaussian with parameter $\sigma_G = \sqrt{\frac{I}{4} + \frac{\sigma_D^2(1+q)}{q}}$ for any $t = 1, 2, \cdots, T$.

Recall $G_t = \sum_{s=1}^{t-1} \mathbb{I}(D_s \geq t-s)$. When $t \leq \mu_D + M_D - 1$, $G_t \leq \mu_D + M_D$. When $t \geq \mu_D + M_D - 1$, specifying $k_i = t - (\mu_D + M_D) - i$ and $\tilde{D}_{k_i} = D_i - \mu_D - M_D$,

$$
\begin{aligned}
G_t &= \sum_{s=1}^{t-1} \mathbb{I}(D_s \geq t - s) \\
&= \sum_{s=1}^{t-\mu_D-M_D-1} \mathbb{I}(D_s \geq t - s) + \sum_{s=t-\mu_D-M_D}^{t-1} \mathbb{I}(D_s \geq t - s) \\
&= \sum_{s=t-\mu_D-M_D}^{t-1} \mathbb{I}(D_s \geq t - s) + \sum_{s=1}^{t-\mu_D-M_D-1} \mathbb{I}(D_s - \mu_D - M_D \geq t - s - \mu_D - M_D) \\
&\leq \mu_D + M_D + \sum_{s=1}^{t-\mu_D-M_D-1} \mathbb{I}(D_s - \mu_D - M_D \geq t - s - \mu_D - M_D) \\
&= \mu_D + M_D + \sum_{i=1}^{t-\mu_D-M_D-1} \mathbb{I}(D_{t-(\mu_D+M_D)-i} - \mu_D - M_D \geq i) \quad (i = t - s - \mu_D - M_D) \\
&= \mu_D + M_D + \sum_{i=1}^{t-\mu_D-M_D-1} \mathbb{I}(\tilde{D}_{k_i} \geq i)
\end{aligned}
$$

Hence,

$$
\begin{aligned}
G_t &\leq \sum_{i=1}^{t-\mu_D-M_D-1} [\mathbb{I}(\tilde{D}_{k_i} \geq i) - p_i] + (\sum_{i=1}^{t-\mu_D-M_D-1} p_i) + \mu_D + M_D \\
&= \mu_D + M_D + \sum_{i=1}^{t-\mu_D-M_D-1} \tilde{I}_i + (\sum_{i=1}^{t-\mu_D-M_D-1} p_i) \\
&\leq \mu_D + M_D + \sum_{i=1}^{t-\mu_D-M_D-1} \tilde{I}_i + (\mu_D + M_D) \\
&= \tilde{G}_{t-\mu_D-M_D-1} + 2(\mu_D + M_D)
\end{aligned}
\tag{18}
$$

Therefore, we arrive at $G_t \leq \tilde{G}_{t-\mu_D-M_D-1} + 2(\mu_D + M_D)$ with specific choice of $k_i = t - (\mu_D + M_D) - i$ and $\tilde{D}_{k_i} = D_i - \mu_D - M_D$.

Given the fact that $\mathbb{E}[\tilde{G}_t] = 0$ and $\tilde{G}_t$ is sub-Gaussian with parameter $\sigma_G$, $G_t$ satisfies

$$
\mathbb{P}\left(G_t \geq 2(\mu_D + M_D) + x\right) \leq \exp\left(\frac{-x^2}{2\sigma_G^2}\right).
\tag{19}
$$

**Property 2.** Further define $\tilde{G}_T^* = \max_{1 \leq t \leq T}\{\tilde{G}_t\}$ as the running maximum of correlated sub-exponentials $\tilde{G}_t$ up to time $T$, from Theorem 8, we have

$$
\mathbb{E}[\tilde{G}_T^*] \leq \sigma_G \sqrt{2 \log T}.
$$

By the union bound,

$$
\begin{aligned}
\mathbb{P}\left(\tilde{G}_T^* \geq \sigma_G\sqrt{2\log T} + x\right) &\leq \sum_{t=1}^{T}\mathbb{P}\left(\tilde{G}_t \geq \sigma_G\sqrt{2\log T} + x\right)\\
&\leq T\exp\left(-\frac{(\sigma_G\sqrt{2\log T} + x)^2}{2\sigma_G^2}\right)\\
&= T\exp\left(-\frac{x^2}{2\sigma_G^2} - \frac{2x\sigma_G\sqrt{2\log T}}{2\sigma_G^2} - \log T\right)\\
&= \exp\left(-\frac{x^2}{2\sigma_G^2} - \frac{2x\sigma_G\sqrt{2\log T}}{2\sigma_G^2}\right)\\
&\leq \exp\left(-\frac{x^2}{2\sigma_G^2}\right).
\end{aligned}
$$

Therefore, with probability $1 - \delta$,

$$
\tilde{G}_T^* \leq \sigma_G\sqrt{2\log(T)} + \sigma_G\sqrt{2\log\left(\frac{1}{\delta}\right)}.
$$

Recall that $G_T^* = \max_{1\leq t\leq T} G_t$. When $T \leq \mu_D + M_D - 1$, $G_T^* \leq \mu_D + M_D$. When $T \geq \mu_D + M_D - 1$, specifying $k_i = T - (\mu_D + M_D) - i$ and $\tilde{D}_{k_i} = D_{k_i} - \mu - M$, we have

$$
G_T^* \leq \tilde{G}_T^* + 2(\mu_D + M_D).
$$

The derivation is similar to the analysis in (18).

Therefore, with probability $1 - \delta$, we have

$$
G_T^* \leq 2(\mu_D + M_D) + \sigma_G\sqrt{2\log(T)} + \sigma_G\sqrt{2\log\left(\frac{1}{\delta}\right)}.
$$

**Property 3.** Given a fixed $G_t$ ($t = 1, 2, \cdots, T$), from Vershynin (2010) and Li et al. (2017), $\lambda_{\min}(W_t) \geq B$ with probability $1 - \delta$, when

$$
t \geq \left(\frac{C_1\sqrt{d} + C_2\sqrt{\log(\frac{1}{\delta})}}{\lambda_{\min}(\Sigma)}\right)^2 + \frac{2B}{\lambda_{\min}(\Sigma)} + G_t. \tag{20}
$$

Combining above with (19), we have the desired result.

$\square$

*Proof of Theorem 2.* We first bound the one-step regret. To do so, fix $t$ and let $X_t^* = x_{t,a_t^*}$ and $\Delta_t = \hat{\theta}_t - \theta^*$, where $a_t^* = \arg\max_{a\in[K]} \mu(x_{t,a}'\theta^*)$ is an optimal action at round $t$. The selection of $a_t$ in DUCB-GLCB implies

$$
\langle X_t^*, \hat{\theta}_t\rangle + \beta_t\|X_t^*\|_{V_t^{-1}} \leq \langle X_t, \hat{\theta}_t\rangle + \beta_t\|X_t\|_{V_t^{-1}}.
$$

Then we have

$$
\begin{aligned}
\langle X_t^*, \theta^*\rangle - \langle X_t, \theta^*\rangle &= \langle X_t^* - X_t, \hat{\theta}_t\rangle - \langle X_t^* - X_t, \hat{\theta}_t - \theta^*\rangle \tag{21}\\
&\leq \beta_t(\|X_t\|_{V_t^{-1}} - \|X_t^*\|_{V_t^{-1}}) + \|X_t^* - X_t\|_{V_t^{-1}}\|\Delta\|_{V_t}. \tag{22}
\end{aligned}
$$

Therefore, to bound $\langle X_t^*, \theta^*\rangle - \langle X_t, \theta^*\rangle$, it suffices to bound $\|\Delta\|_{V_t}$ and $\|X_t\|_{V_t^{-1}}$.

Suppose $\lambda_{\min}(W_{\tau+1}) \geq 1$, for any $\delta \in [\frac{1}{T}, 1)$ define event

$$
\mathcal{E}_\Delta := \left\{\|\Delta\|_{W_t} \leq \frac{\sigma}{\kappa}\sqrt{\frac{d}{2}\log\left(1 + \frac{2(t - G_t)}{d}\right) + \log\left(\frac{1}{\delta}\right)}\right\}.
$$

From Lemma 2 in (Li et al. (2017)), then event $\mathcal{E}_\Delta$ holds for all $t \geq \tau$ with probability at least $1 - \delta$.

$$
\begin{aligned}
\|\Delta_t\|_{V_t}^2 &= \Delta_t' V_t \Delta_t = \Delta_t' \left( W_t + \sum_{s \in M_t} X_s X_s' \right) \Delta_t \\
&= \Delta_t' W_t \Delta_t + \sum_{s \in M_t} \Delta_t' X_s X_s' \Delta_t \\
&\leq \Delta_t' W_t \Delta_t + \sum_{s \in M_t} \|\Delta_s\|^2 \|X_s\|^2 \\
&\leq \|\Delta_t\|_{W_t}^2 + G_t \|\Delta_t\|^2.
\end{aligned}
$$

When $\lambda_{\min}(W_t) \geq 16\sigma^2 \frac{d + \log(\frac{1}{\delta})}{\kappa^2}$, from Lemma 7 in (Li et al. (2017)), with probability $1 - \delta$,

$$
\|\Delta_t\|^2 \leq \frac{4\sigma}{\kappa} \sqrt{\frac{d + \log(\frac{1}{\delta})}{\lambda_{\min}(W_t)}} \leq 1.
$$

Therefore, when $\lambda_{\min}(W_t) \geq 16\sigma^2 \frac{d + \log(\frac{1}{\delta})}{\kappa^2}$, with probability $1 - 2\delta$,

$$
\begin{aligned}
\|\Delta_t\|_{V_t} &\leq \sqrt{\frac{\sigma^2}{\kappa^2} \left( \frac{d}{2} \log\left( 1 + \frac{2(t - G_t)}{d} \right) + \log\left( \frac{1}{\delta} \right) \right) + G_t} \\
&\leq \frac{\sigma}{\kappa} \sqrt{\frac{d}{2} \log\left( 1 + \frac{2(t - G_t)}{d} \right) + \log\left( \frac{1}{\delta} \right)} + \sqrt{G_t}. \tag{23}
\end{aligned}
$$

Let us come back to the satisfaction of conditions $\lambda_{\min}(W_t) \geq 16\sigma^2 \frac{d + \log(\frac{1}{\delta})}{\kappa^2}$ and $\lambda_{\min}(W_{\tau+1}) \geq 1$. From Proposition 1, $\lambda_{\min}(W_t) \geq \max\left\{ 1, 16\sigma^2 \frac{d + \log(\frac{1}{\delta})}{\kappa^2} \right\}$ with probability $1 - 2\delta$, when

$$
t \geq \left( \frac{C_1\sqrt{d} + C_2\sqrt{\log(\frac{1}{\delta})}}{\lambda_{\min}(\Sigma)} \right)^2 + \frac{2\max\{1, 16\sigma^2 \frac{d + \log(\frac{1}{\delta})}{\kappa^2}\}}{\lambda_{\min}(\Sigma)} + 2(\mu_D + M_D) + \sigma_G \sqrt{2\log\left( \frac{1}{\delta} \right)} := \tau. \tag{24}
$$

We now choose $\beta_t = \frac{\sigma}{\kappa} \sqrt{\frac{d}{2} \log\left( 1 + \frac{2(t - G_t)}{d} \right) + \log(\frac{1}{\delta})} + \sqrt{G_t}$. If $\mathcal{E}_t$ holds for all $t \geq \tau$, then,

$$
\langle X_t^*, \theta^* \rangle - \langle X_t, \theta^* \rangle \leq \beta_t \left( \|X_t\|_{V_t^{-1}} - \|X_t^*\|_{V_t^{-1}} + \|X_t^* - X_t\|_{V_t^{-1}} \right). \tag{25}
$$

Suppose there is an integer $m$ such that $\lambda_{\min}(V_{m+1}) \geq 1$, from Lemma 2 in Li et al. (2017), we have

$$
\sum_{t=m+1}^{m+n} \|X_t\|_{V_t^{-1}} \leq \sqrt{2dn \log\left( \frac{n + m}{d} \right)}. \tag{26}
$$

for all $n \geq 0$. Combine (25) and (26), we have

$$
\begin{aligned}
\sum_{t=\tau+1}^{T} \left( \langle X_t^*, \theta^* \rangle - \langle X_t, \theta^* \rangle \right) &\leq 2 \max_{1 \leq t \leq T} \{\beta_t\} \sqrt{2Td \log\left( \frac{T}{d} \right)} \\
&\leq 2 \left[ \frac{\sigma}{\kappa} \sqrt{\frac{d}{2} \log\left( 1 + \frac{2T}{d} \right) + \log\left( \frac{1}{\delta} \right)} + \sqrt{G_T^*} \right] \sqrt{2Td \log\left( \frac{T}{d} \right)} \\
&\leq 2\sqrt{G_T^*} \sqrt{2Td \log\left( \frac{T}{d} \right)} + \frac{2d\sigma}{\kappa} \log\left( \frac{T}{d\delta} \right) \sqrt{T}.
\end{aligned}
$$

Note that $g$ is an increasing Lipschitz function with Lipschitz constant $L_g$ and the $g$ function is bounded between 0 and 1. The regret of algorithm DUCB-GLCB can be upper bounded as

$$
\begin{aligned}
R_T &\leq \tau + L_g \sum_{t=\tau+1}^{T} \left( \langle X_t^*, \theta^* \rangle - \langle X_t, \theta^* \rangle \right) \\
&\leq \tau + L_g \left( 2\sqrt{G_T^*} \sqrt{2Td\log\left(\frac{T}{d}\right)} + \frac{2d\sigma}{\kappa} \log\left(\frac{T}{d\delta}\right) \sqrt{T} \right).
\end{aligned}
\tag{27}
$$

Combining with the results in (6), (23) and (24), with probability $1 - 5\delta$,

$$
\begin{aligned}
R_T &\leq \tau + L_g \left[ 2 \left( \sqrt{2(\mu_D + M_D)} + \sqrt{\sigma_G}(2\log(T))^{1/4} + \sqrt{\sigma_G}\left(2\log\left(\frac{1}{\delta}\right)\right)^{1/4} \right) \sqrt{2Td\log\left(\frac{T}{d}\right)} \right. \\
&\qquad \left. + \frac{2d\sigma}{\kappa} \log\left(\frac{T}{d\delta}\right) \sqrt{T} \right] \\
&= \tau + L_g \left[ 4\sqrt{\mu_D + M_D} \sqrt{Td\log\left(\frac{T}{d}\right)} + 2^{7/4} \sqrt{\sigma_G} \left(\log\left(\frac{1}{\delta}\right)\right)^{1/4} \sqrt{d\log\left(\frac{T}{d}\right) T} \right. \\
&\qquad \left. + 2^{7/4} \sqrt{\sigma_G} (\log(T))^{1/4} \sqrt{d\log\left(\frac{T}{d}\right) T} + \frac{2d\sigma}{\kappa} \log\left(\frac{T}{d\delta}\right) \sqrt{T} \right].
\end{aligned}
$$

$\square$

*Proof of Proposition 4.* When there exists an upper bound $D_{\max}$ on the delay, Proposition 1 can be improved as follows.

Then there exist positive, universal constants $C_1$ and $C_2$ such that $\lambda_{\min}(W_t) \geq B$ with probability at least $1 - \delta$, as long as

$$
t \geq \left( \frac{C_1 \sqrt{d} + C_2 \sqrt{\log(\frac{1}{\delta})}}{\lambda_{\min}(\Sigma)} \right)^2 + \frac{2B}{\lambda_{\min}(\Sigma)} + D_{\max}.
$$

Along with the fact that event (23) holds for all $t \geq \tau$ with probability at least $1 - 2\delta$, we have with probability $1 - 3\delta$,

$$
(27) \leq \tau + L_g \left( 2\sqrt{D_{\max}} \sqrt{2Td\log\left(\frac{T}{d}\right)} + \frac{2d\sigma}{\kappa} \log\left(\frac{T}{d\delta}\right) \sqrt{T} \right).
$$

That is, $O(R_T) = O(\sqrt{D_{\max}}\sqrt{dT\log(T)} + d\sqrt{T}\log(T))$

When $\{D_t\}_{t=1}^{T}$ are **iid** with mean $\mu_D'$,

$$
\begin{aligned}
\mathbb{E}[G_t] &= \mathbb{E}\left[\sum_{s=1}^{t-1} \mathbb{I}(s + D_s \geq t)\right] = \sum_{s=1}^{t-1} \mathbb{P}(s + D_s \geq t) \leq \mu_D', \\
\mathbb{V}[G_t] &= \mathbb{V}\left[\sum_{s=1}^{t-1} \mathbb{I}(s + D_s \geq t)\right] \leq \sum_{s=1}^{t-1} \mathbb{P}(s + D_s \geq t) \leq \mu_D'.
\end{aligned}
$$

Therefore, with probability $1 - 5\delta$,

$$
\begin{aligned}
(27) \leq \tau + L_g \left[ 4\sqrt{\mu_D'} \sqrt{Td\log\left(\frac{T}{d}\right)} + 2^{7/4} \sqrt{\sigma_G} \left(\log\left(\frac{1}{\delta}\right)\right)^{1/4} \sqrt{d\log\left(\frac{T}{d}\right) T} \right. \\
\left. + 2^{7/4} \sqrt{\sigma_G} (\log(T))^{1/4} \sqrt{d\log\left(\frac{T}{d}\right) T} + \frac{2d\sigma}{\kappa} \log\left(\frac{T}{d\delta}\right) \sqrt{T} \right].
\end{aligned}
$$

$\square$

*Proof of Lemma 6.* Define

$$
\begin{aligned}
s_{t,a} &= \sqrt{x_{t,a}^T A_t^{-1} x_{t,a}} \in \mathbb{R}_+ \\
B_t &= [x_{\tau,a_\tau}^T]_{\tau \in \Psi_t} \in \mathbb{R}^{|\Psi_t| \times d} \\
C_t &= [\mathbb{I}(D_\tau + \tau < t - 1) x_{\tau,a_\tau}^T]_{\tau \in \Psi_t} \in \mathbb{R}^{|\Psi_t| \times d} \\
Z_t &= [y_{\tau,a_\tau}]_{\tau \in \Psi_t} \in \mathbb{R}^{|\Psi_t| \times 1}.
\end{aligned}
$$

Then $A_t = I_d + B_t^T B_t$ and $c_t = C_t^T Z_t$. (Note that $A_t$ and $c_t$ are defined in Algorithm 2.)

$$
\begin{aligned}
\hat{y}_{t,a} - x_{t,a}' \theta^* &= x_{t,a}' \theta_t - x_{t,a}' \theta^* \\
&= x_{t,a}' A_t^{-1} c_t - x_{t,a}' A_t^{-1} (I_d + B_t' B_t) \theta^* \\
&= x_{t,a}' A_t^{-1} C_t' Z_t - x_{t,a}' A_t^{-1} (\theta^* + B_t' B_t \theta^*) \\
&= x_{t,a}' A_t^{-1} B_t' (Z_t - B_t \theta^*) + x_{t,a}' A_t^{-1} (C_t - B_t)' Z_t - x_{t,a}' A_t^{-1} \theta^*.
\end{aligned}
$$

Since $\|\theta^*\| \le 1$,

$$
|\hat{y}_{t,a} - x_{t,a}' \theta^*| \le |x_{t,a}' A_t^{-1} B_t' (Z_t - B_t \theta^*)| + \|x_{t,a}' A_t^{-1}\| \|(C_t - B_t)' Z_t\| + \|x_{t,a}' A_t^{-1} \theta^*\|.
$$

Due to the statistical independence of samples indexed in $\Psi_t$, we have $\mathbb{E}[Z_t - B_t \theta^*] = 0$. Denote $\bar{\alpha} = \sqrt{\frac{1}{2} \ln\left(\frac{2TK}{\delta}\right)}$, following the analysis in (Chu et al., 2011, Lemma 1), we have

$$
\mathbb{P}(|x_{t,a}' A_t^{-1} B_t' (Z_t - B_t \theta^*)| > \bar{\alpha} s_{t,a}) \le 2 \exp\left(-\frac{2\bar{\alpha}^2 s_{t,a}^2}{\|B_t A_t^{-1} x_{t,a}\|^2}\right) \le 2 \exp\left(-2\bar{\alpha}^2\right) = \frac{\delta}{TK},
$$

and $\|A_t^{-1} x_{t,a}\| \le s_{t,a}$.

Further notice that $\|(B_t - C_t)' Z_t\| \le G_t$. Combining above facts, we arrive at the desired result. $\square$

*Proof of Lemma 7.* By Lemma 3 in Chu et al. (2011), for any $s \in [S]$,

$$
\sum_{\tau \in \Psi_{T+1}^s} s_{\tau,a_\tau} \le 5\sqrt{d|\Psi_{T+1}^s| \log |\Psi_{T+1}^s|}.
$$

Hence,

$$
\begin{aligned}
\sum_{\tau \in \Psi_{T+1}^s} w_{\tau,a_\tau} &= \sum_{\tau \in \Psi_{T+1}^s} \alpha_\tau s_{\tau,a_\tau} \\
&\le 5(\bar{\alpha} + G_T^* + 1)\sqrt{d|\Psi_{T+1}^s| \log |\Psi_{T+1}^s|} \\
&\le 5\sqrt{2}\bar{\alpha}(\bar{\alpha} + G_T^* + 1)\sqrt{d|\Psi_{T+1}^s|}.
\end{aligned} \tag{28}
$$

(28) holds since $\sqrt{2}\bar{\alpha} \ge \sqrt{\log T} \ge \sqrt{\log |\Psi_{T+1}^s|}$. On the other hand, by Step 13 of Algorithm 3 (SupLinUCB) in Chu et al. (2011),

$$
\sum_{\tau \in \Psi_{T+1}^s} w_{\tau,a_\tau} \ge 2^{-s} |\Psi_{T+1}^s|. \tag{29}
$$

Therefore,

$$
|\Psi_{T+1}^s| \le 2^s 5\sqrt{2}\bar{\alpha}(\bar{\alpha} + G_T^* + 1)\sqrt{d|\Psi_{T+1}^s|}
$$

$\square$

*Sketch proof of Theorem 5.* Denote $\Phi_0$ be the set of trails for which an alternative is chosen in step 7-8 of Algorithm 3. Since $2^{-S} \leq \frac{1}{\sqrt{T}}$ we have $\{1, 2, \cdots, T\} = \Phi_0 \cup_s \Phi_{T+1}^s$.

$$
\begin{aligned}
\mathbb{E}\left[R_T\right] &= \sum_{t=1}^{T}\left[\mathbb{E}<X_t^*, \theta^*>-\mathbb{E}<X_t, \theta^*>\right] \\
&= \sum_{t \in \Phi_0}\left[\mathbb{E}<X_t^*, \theta^*>-\mathbb{E}<X_t, \theta^*>\right]+\sum_{s=1}^{S} \sum_{t \in \Phi_{t+1}^s}\left[\mathbb{E}<X_t^*, \theta^*>-\mathbb{E}<X_t, \theta^*>\right] \\
&\leq \frac{2}{\sqrt{T}}|\Phi_0|+\sum_{s=1}^{S} 8 \cdot 2^{-s}|\Phi_{T+1}^s| \quad (30) \\
&\leq \frac{2}{\sqrt{T}}|\Phi_0|+\sum_{s=1}^{S} 40(\sqrt{2} \bar{\alpha}(G_T^*+\bar{\alpha}+1)) \sqrt{d|\Phi_{T+1}^s|} \quad (31) \\
&\leq 2 \sqrt{T}+40(\sqrt{2} \bar{\alpha}(G_T^*+\bar{\alpha}+1)) \sqrt{S T d} \quad (32)
\end{aligned}
$$

with probability $1-\delta S$. (30) holds by (Auer, 2002, Lemma 15) or (Chu et al., 2011, Lemma 5), (31) holds by Lemma 7, and (32) holds by some simple calculations.

Apply the Azuma-Hoeffding bound (Auer, 2002, Lemma 8) with $\alpha_\tau=2$ and $B=4\sqrt{T \log\left(\frac{2}{\delta}\right)}$, we have

$$
R_T \leq 2 \sqrt{T}+46\left(\sqrt{2} \bar{\alpha}(G_T^*+\bar{\alpha}+1)\right) \sqrt{S T d}, \quad (33)
$$

with probability $1-\delta(S+1)$. Recall that $\bar{\alpha}=\sqrt{\frac{1}{2} \ln\left(\frac{2 T K}{\delta}\right)}$. Replacing $\delta$ by $\delta/(S+1)$, substituting $S=\log(T)$, and combining with the result in (6) yields

$$
\begin{aligned}
R_T \leq \quad & 2 \sqrt{T}+46 \sqrt{\log(T) T d}\left(\sqrt{2} \sqrt{\frac{1}{2} \log\left(\frac{2 T K(\log(T)+1)}{\delta}\right)}\left(2(\mu_D+M_D)\right.\right. \\
& + \sigma_G \sqrt{2 \log(T)}+\sigma_G \sqrt{2 \log\left(\frac{1}{\delta}\right)}+\sqrt{\frac{1}{2} \log\left(\frac{2 T K(\log(T)+1)}{\delta}\right)}+1\bigg)\bigg)
\end{aligned}
$$

with probability $1-2\delta$. $\qquad \square$