[Reviews · NeurIPS 2019]

Reviewer 1



This paper studies online learning in the generalized linear contextual bandits where rewards are observed with delays. UCB-based and Thompson sampling based algorithms are proposed, respectively. Theoretical analysis of these two algorithm in terms of regret bounds are provided. The problem is very fundamental and practical. The proposed algorithms have theoretical perform and the theoretical analysis is reasonable. It may be interesting to consider some real applications and test the performance of the proposed algorithms.

Reviewer 2



BRIEF SUMMARY ------------------------- In this work, authors study the GLB problem with delayed feedback. In this setting, they describe how to adapt the UCB and TS (bayesian) algorithms for which they prove similar frequentist and Bayesian regret bounds respectively. They work under the assumption of independent stochastic delays, but they allow the distribution to have a near-heavy tail distribution, hence going beyond the sub-Gaussian case. Finally, they also provide a characterization of the regret under more restrictive assumptions on the delays (iid or bounded). GENERAL REMARKS ------------------------------ The paper is well written and easy to follow. The problem is well motivated and I appreciated that they discuss the regret bound they obtain w.r.t. the structure of the delay. In particular, how the regret bounds simplify when the delays are iid or bounded brings insight as it allows a straightforward comparison to previous work. On the negative side, the algorithm involves some universal (non-explicit) constant which impacts the tuning parameter $\tau$. It would be nice to have a discussion on how to set $\tau$ appropriately. Further, the technical contribution consists mainly in adapting existing result on delayed feedback and GLB regret bounds. That being said, I believe this is offset by the novelty (to the best of my knowledge) of the result and because it brings a valuable contribution to the contextual bandit literature. COMMENTS AND QUESTIONS ------------------------------------ 1) TS versus UCB. The comparison between the two algorithm is somehow unfair. Although it is commonly acknowledge that TS performs better than UCB in practice and is easier to implement, it is mainly because the DTS instance considered here is design for the **Bayesian regret** guarantee. The TS algorithm with **frequentist regret** guarantee in GLB (see [1]) requires a similar careful design of the confidence parameter as UCB and moreover offers a worse regret bound by $\sqrt{d}$. To compare the two algorithms, one should consider the instances which offer the same type of guarantee (Frequentist or Bayesian). The frequentist guarantee is way stronger than the Bayesian one (it actually implies it), which explains why DUCB is more complicated than DTS. The consequence is that Bayesian TS is way less aggressive in the exploration than UCB which may explain its better empirical performances. As a result, I would be cautious in doing such comparison (in Sec. 1.2 and Conclusion), unless experiments are provided and conducted with the appropriate algorithm instances. 2) Arm set. Authors consider the finite arm case, where $K$ arms are sample iid at each time step from some distribution which should satisfy that they span (in some sense) every directions. What about the infinite arm case ? Regret bounds have been derived in this setting for both UCB and TS, and I believe that this analysis can extend to this case. Is it ? What if the arms are fixed at the beginning of the trajectory ? Again it seems that it would only require modifying Asm. 1. Overall, I think that the structure of the arm set can be relaxed with minor modifications, which would allow for a less restricted setting. MINOR COMMENTS ------------------------------------ - Check the numbering of Thm, Prop and Cor in the main text. - In the related work section, you may also mention [1] and [2] for the TS in GLB and Bandit with delay respectively. - Line 207. $p$ should be $q$ isn't it ? REFERENCES ------------------- [1] M. Abeille, A Lazaric. Linear Thompson sampling revisited - Electronic Journal of Statistics, 2017 [2] Vernade, C., CappĂ©, O., & Perchet, V. (2017). Stochastic bandit models for delayed conversions.

Reviewer 3



The problem considered in this paper is well motivated, and the constraint of delay is of practical interest. Also, it looks like this is the first paper that extends existing literature into the stochastic parametric bandit setting. Two things I would like to see, but do not appear are: 1. A lower bound for learning under stochastic delay. Although I understand that the regret bounds provided here collapse to the optimal regret bound when no delay exists, it would still be great if one can fully characterize the hardness of the problem. 2. A more interesting DTS analysis would go into the frequentist framework similar to [1]. Even for the Bayes setting described here, it would be helpful if the authors could describe in details what is the unique difficulty of extending the results from DUCB to DTS if one simply wants to adopt the reduction framework proposed in [2]. [1] S. Agrawal, N. Goyal, "Thompson Sampling for contextual bandits with linear payoffs". In ICML 2013. [2] Russo D, Van Roy B, "Learning to optimize via posterior sampling". in Mathematics of Operations Research 2014.

[Author Response · NeurIPS 2019]

We thank all three reviewers for their careful readings, valuable questions and constructive suggestions. We will revise the paper thoroughly and incorporate all the comments.

**[reviewer 1]** Thanks for the suggestion on adding a section of experiments. We agree that numerical results can enrich our paper and make it more convincing from an empirical perspective. And we are certainly happy to run some simulations to show the performance of our algorithms and to compare DUCB and DTS algorithms.

**[reviewer 2]** Thanks for raising the questions on infinite arms and frequentist regret bound for DTS. These are all great directions to explore and to further improve our paper. Thanks for the related references and we will add them in the revision.

(1) **DUCB versus DTS**: Thank you for raising this point, which we did not make clear in the initial submission! The goal of our paper is not to compare the frequentist regret bound for UCB with bayesian regret bound for DTS, but rather to characterize the regret performances in these two regimes for the two algorithms. Yes we agree that the frequentist bound for DTS can be derived by adapting [2, 1] for which it may need more modifications. This is an interesting direction to explore and we will definitely keep it in mind. The frequentist bound for DTS is stronger than the bayesian regret bound. We agree with the reviewer that this is exactly the reason why the frequentist bound for DUCB looks more complicated than the bayesian regret bound for DTS. Thanks for raising this point and we will add a remark on it. In addition, we are certainly happy to run some simulations to show the performance of our algorithms and to compare DUCB and DTS algorithms.

(2) **Infinite arms**: Our analysis does not depend on the assumption that the arms are finite. It also works with infinite arms. Note that the regret bound in our paper is not tight in terms of the feature dimension $d$. Our goal is to provide some clean and insightful analysis in simple settings. We can further tighten the regret bound from $d$ to $\sqrt{d}$ by assuming the number of arms is finite and possibly changing and by using the baselinUCB/suplinUCB decomposition. (See [3] for the case without delays. Also, when the number of arms is both finite and fixed, the $O(\sqrt{dT})$ bound can be achieved with a simpler analysis.) We will add a remark on this point in the revision.

(3) **How to set** $\tau$: In Algorithm 1, the warm-up period $\tau$ is to ensure that $\mathbb{E}[\sum_{t=1}^{\tau} X_t X_t'] \geq 1$. By Proposition 1 (equation (7)),

$$\tau = \left( \frac{C_1 \sqrt{d} + C_2 \sqrt{\log\left(\frac{1}{\delta}\right)}}{\lambda_{\min}(\Sigma)} \right)^2 + \frac{2B}{\lambda_{\min}(\Sigma)} + 2(\mu_D + M_D) + \sigma_G \sqrt{2 \log\left(\frac{1}{\delta}\right)}. \tag{1}$$

The RHS of (1) is determined by the model parameters and can be directly calculated. Note that $\tau = O\left(d + \log\left(\frac{1}{\delta}\right)\right)$, i.e. the larger the feature dimension $d$ is, the longer the warm-up period $d$ is. This is not surprising since the algorithm needs a long time to gather information when the feature dimension is high. Also, $\tau$ increases as $\delta$ decreasing, which implies that it takes a longer warm-up period when the confidence level $1 - \delta$ is larger. $\tau$ in Theorem 5 can be set similarly. We will add a remark on this point in the revision.

(4) **Numbering issue**: Thanks for catching it. We will fix the numbering issue in the revision.

(5) **References**: Thanks for the references ([4, 1]). We will add them in the revision.

(6) **Typos**: Thanks for catching the typo. Yes it should be $q$ instead of $p$ in line 207. We will change it.

**[reviewer 3]** Thanks for raising the questions on the lower bound and frequentist regret bound for DTS. These are all great directions to explore and to further improve our paper.

(1) **Frequentist regret bound for DTS**: The frequentist regret bound for DTS is stronger than the bayesian regret bound. We agree with the reviewer that the frequentist bound can be derived by adapting [2]. This is a great future direction to explore and we will definitely keep it in mind.

(2) **Lower bound**: The lower bound is certainly worth trying. It seems require some extra efforts and we will explore it in the future.

# References

[1] Marc Abeille, Alessandro Lazaric, et al. Linear thompson sampling revisited. *Electronic Journal of Statistics*, 11(2):5165–5197, 2017.

[2] Shipra Agrawal and Navin Goyal. Thompson sampling for contextual bandits with linear payoffs. In *International Conference on Machine Learning*, pages 127–135, 2013.

[3] Lihong Li, Yu Lu, and Dengyong Zhou. Provably optimal algorithms for generalized linear contextual bandits. In *Proceedings of the 34th International Conference on Machine Learning-Volume 70*, pages 2071–2080. JMLR. org, 2017.

[4] Claire Vernade, Olivier Cappé, and Vianney Perchet. Stochastic bandit models for delayed conversions. *arXiv preprint arXiv:1706.09186*, 2017.


[Meta-Review · NeurIPS 2019]

Reviewers appreciated the work's nod to practicality to incorporate delays and generalized linear model. A well-prepared paper.